# Detection of Preclinical Orthostatic Disorders in Young African and European Adults Using the Head-Up Tilt Test with a Standardized Hydrostatic Column Height: A Pilot Study

**DOI:** 10.3390/biomedicines10092156

**Published:** 2022-09-01

**Authors:** Victor N. Dorogovtsev, Dmitry S. Yankevich, Marina V. Petrova, Vladimir I. Torshin, Aleksander E. Severin, Ilya V. Borisov, Julia A. Podolskaya, Andrey V. Grechko

**Affiliations:** 1Federal Research and Clinical Center of Intensive Care Medicine and Rehabilitology, 107031 Moscow, Russia; 2Department of Anesthesiology and Resuscitation with Medical Rehabilitation Courses, Peoples’ Friendship University of Russia (RUDN University), 6 Miklukho-Maklaya Street, 117198 Moscow, Russia; 3Department of Normal Physiology, and Department of Anesthesiology and Resuscitation with Medical Rehabilitation Courses, Peoples’ Friendship University of Russia (RUDN University), 6 Miklukho-Maklaya Street, 117198 Moscow, Russia

**Keywords:** racial differences, head-up tilt test, hydrostatic column height standardization, blood pressure, hemodynamics, preclinical orthostatic disorders, vascular stiffness

## Abstract

Arterial hypertension (AH) remains the most common disease. One possible way to improve the effectiveness of the primary prevention of AH is to identify and control the preclinical orthostatic disturbances that precede the development of AH. The aim of the study was to determine the feasibility of a new protocol for the head-up tilt test (HUTT) with a standardized hydrostatic column height for the detection of asymptomatic orthostatic circulatory disorders and their racial differences in young African and European adults. Methods. In total, 80 young healthy adults (40 African and 40 European) aged 20–23 years performed the HUTT with a standardized hydrostatic column height of 133 cm. The hemodynamic parameters were recorded using a Task Force Monitor (3040i). The cardio-ankle vascular index (CAVI) was measured using a VaSera VS-2000 volumetric sphygmograph. Results. The baseline and orthostatic hemodynamic changes in both racial groups were within normal limits. Orthostatic circulatory disturbances were not detected in 70% of the European participants and 65% of the African participants; however, preclinical orthostatic hypertension, which precedes AH, was detected using the new HUTT protocol in 32.5% of the African participants and 20% of the European participants. The baseline CAVI was higher in the European group compared to the African group. Conclusion. The results of this study showed the feasibility of the detection of preclinical orthostatic disturbances in young adults and the detection of their racial differences using the HUTT protocol, providing the use of a standard gravity load. Further study on the evolution of preclinical orthostatic disturbances and their relation to increased vascular stiffness is necessary among large samples.

## 1. Introduction

Arterial hypertension (AH) remains one of the most common diseases worldwide [1,2]. The highest prevalence of AH is found in African countries [3]. AH is the most important risk factor for cardiovascular diseases (CVDs) [4,5] and is associated with an increased risk of all-cause mortality [6]. AH does not begin with a rise in blood pressure (BP) and factors that significantly increase the risk of developing AH have been identified in healthy young populations. This determines the relevance of research that aims to identify early preclinical abnormalities, which may have pathogenetic links to the development of AH.

Current recommendations for the primary prevention of hypertension involve a population-based approach, which includes engaging in moderate physical activity, maintaining normal body weight, limiting alcohol consumption and smoking, reducing sodium intake, maintaining an adequate intake of potassium, and consuming a diet that is rich in fruits, vegetables, and low-fat dairy products [7,8,9]. The global arterial hypertension pandemic is an indicator of the lack of effectiveness among the existing AH primary prevention strategies; therefore, new ways to improve the efficiency of health systems have recently been proposed [10]. New sensitive tools are required to detect early preclinical signs of AH and assess the individual effectiveness of preventive measures.

Orthostatic hypotension (OH) and hypertension (OHT) precede AH and other CVDs [11,12]. The criterion for the diagnosis of OH is a decrease in systolic blood pressure (SBP) during orthostasis compared to horizontal position (ΔSBP) ≤ −20 mmHg and the criterion for the diagnosis of OHT is an increase in ΔSBP ≥ +20 mmHg [13,14]. Even less significant orthostatic disturbances in healthy young adults and older adults without CVDs can significantly increase the risks of AH [15,16]. These data suggest that OH and OHT may be the result of the evolution of asymptomatic orthostatic abnormalities, which are even seen in the young healthy population.

The importance of the problem determines the need for the accurate detection of the orthostatic disturbances that precede the development of AH. In the above studies, the active standing test was applied [17,18]. A fixed-angle HUTT has also been used to diagnose orthostatic dysfunction [19,20]. The disadvantage of these protocols is that in the upright position, a person’s vascular system is subjected to hydrostatic pressure that is proportional to their height. Significant differences in height determine large variations in hydrostatic column height and pressure, which can have a negative impact on the accurate detection of orthostatic abnormalities.

The aim of this study was to detect preclinical orthostatic abnormalities and their racial differences in young African and European adults using a newly developed head-up tilt test (HUTT) with a standardized height for the hydrostatic column (Luanda protocol) and individual tilt angles [21]. This protocol allowed the results of orthostatic circulatory regulation to be assessed, irrespective of the effect of the individual’s height (hydrostatic pressure). Given the higher prevalence of AH in African countries compared to European countries, it was important to explore possibilities for the early detection of preclinical orthostatic abnormalities [22] and their racial differences using the new HUTT test to assess young healthy European and African adults.

## 2. Materials and Methods

### 2.1. Participants

The inclusion criteria were the following: healthy subjects between 20–23 years of age; a body mass index from 18 to 30 kg/m^2^; a blood pressure of 100–130/75–85 mm Hg; not taking any medication. The participants were advised to abstain from smoking and alcohol consumption for 2 days prior to the study, as well as to abstain from drinking coffee and carrying out intensive exercise for 24 h before the measurements.

The exclusion criteria were as follows: cardiac rhythm disorders; treatment with vasoactive or antiarrhythmic drugs; a history of heart failure or orthostatic intolerance. Participants with orthostatic disorders that were associated with primary and secondary autonomic nervous system dysfunctions or other diseases (such as myelitis, syringomyelia, paraneoplastic syndrome, diabetes, renal failure, anemia, multiple sclerosis, the effects of AH treatment, etc.) were excluded.

In total, 40 healthy young European participants (Group 1) and 40 healthy young African participants (Group 2) without arterial hypertension (AH), atherosclerosis, diabetes mellitus or orthostatic intolerance were included in the study (Table 1).

The study was carried out 2 h after a light breakfast, between 8:00 and 11:30 a.m., in a laboratory with a room temperature between 24 and 25° C, a humidity of 50–55%, and minimal ambient noise. Young women were investigated during their postmenstrual period.

### 2.2. Head-Up Tilt Test (HUTT) Procedure

The HUTT was performed using an electrically operated tilt table and according to the Luanda protocol, which consisted of 3 phases: supine position for 10 min; HUTT position for 10 min; return to supine position for 10 min. This protocol included the setting of an individual tilt angle to establish the standard height of 133 cm for the hydrostatic columns of all participants, irrespective of their height [21].

### 2.3. Hemodynamic Measurements

All measurements were taken using a Task Force Monitor (TFM) (CNSystems Medizintechnik, Graz, Austria). All data were obtained from 6 to 10 min into each of the 3 positions (i.e., supine, HUTT, and return to supine) and then the data were averaged. The systolic blood pressure (SBP) and diastolic blood pressure (DBP) were measured at the right brachial artery using the oscillometric method and we averaged the data that were obtained between 6 and 10 min into each of the 3 stages of the tilt test. The hemodynamic parameters, such as stroke volume (SV), cardiac output (CO), and total peripheral resistance (TPR), were recorded beat-to-beat throughout the study using impedance cardiography [23]. The computer calculation of TPR was carried out using CO and blood pressure values that were measured with a finger sensor using the automatic oscillometric method calibration and the heart rate measurements were carried out using RR intervals.

### 2.4. Cardio-Ankle Vascular Index (CAVI)

The *CAVI* [24] was measured using a VaSera VS-2000 volumetric sphygmograph (Fukuda Denshi Corp., Tokyo, Japan). The vascular stiffness was measured in the supine position at the beginning of the study.

### 2.5. Statistical Analysis

The statistical analysis was carried out using STATISTICA 10 software (Stat Soft^®^), Version 10, Tulsa, OK, USA. The nominal data were described with absolute values. The quantitative data with normal distributions were combined into a variation series, in which the arithmetic mean (M) and standard deviation (SD) were calculated. In the absence of a normal distribution, the quantitative data were presented as the median and interquartile range (25–75%). The Kolmogorov–Smirnov test was used to verify the nature of the distributions. Student’s t-test was used to compare the mean values of normally distributed populations of quantitative data. The Mann–Whitney U test was used to compare two independent groups when there was no evidence of normal distribution. Wilcoxon’s W test was used to analyze the statistical significance of differences in quantitative characteristics between the two dependent samples. Differences were considered statistically significant at *p* < 0.05.

## 3. Results

### 3.1. The Results of the Orthostatic Changes and an Inter-Group Analysis of Selected Hemodynamic Parameters Are Presented in Table 2

The intra-racial differences in the hemodynamic parameters are presented in the upper part of the table for each of the three positions: (I, supine; II, HUTT; III, return to supine position). The significance of the inter-group differences (*p*) in individual parameters is presented below the data from the two groups for each of the three positions (I, II, and III). The significance of the orthostatic changes in the selected parameters for the separate groups are shown at the bottom of the table: *pI-II* depicts the significance of the differences between parameters in the supine and HUTT positions; *pI-III* denotes the significance of the differences between the supine and return to supine positions; *pII-III* represents the significance of the differences between the HUTT and return to supine positions. The baseline (supine) hemodynamic parameters for both groups were within normal limits (Table 2). Group 1 showed higher CO (*p* < 0.001) and SV (*p* = 0.095) values, while the other baseline hemodynamic parameters were higher in Group 2 (Table 2).

**Table 2 biomedicines-10-02156-t002:** The inter-group analysis of selected hemodynamic parameters at different stages of the HUTT.

Racial Identity	Hemodynamic Parameters					
	SBP (mmHg)	DBP (mmHg)	HR (b/min)	SV (mL)	CO (L/min)	TPR (din·sek·sm-5)
I. Supine						
Group 1 (*n =* 40)	109.2 [103.4; 118.7]	68.4 [65.3; 74.0]	66.0 [62.5; 71.2]	92.7 [77.7; 105.3]	6.7 [5.6; 7.6]	998.7 [881.6; 1155.1]
Group 2 (*n =* 40)	115.0 [111.5; 120.6]	71.9 [67.9; 77.3]	65.5 [60; 73.7]	81.6 [75.0; 94.6]	5.5 [4.7; 6.1]	1282.9 [1096.7; 1435.1]
Inter-Group Differences	*p* = 0.022	*p* = 0.056	*p* = 0.9	*p* = 0.095	*p* < 0.001	*p* < 0.001
II. HUTT						
Group 1 (*n* = 40)	111.9 [108.2; 119.6]	74.6 [69.8; 78.8]	80.6 [74.0; 85.0]	66.7 [58.1; 75.7]	5.9 [5.5; 6.7]	1168.7 [1024.1; 1257.0]
Group 2 (*n* = 40)	120.3 [112.4; 123.4]	77.6 [74.1; 82.8]	76.2 [67.3; 88.3]	70.9 [64.0; 79.0]	5.6 [5.0; 6.3]	1304.4 [1116.8; 1504.2]
Inter-Group Differences	*p* < 0.001	*p* = 0.014	*p* = 0.25	*p* = 0.11	*p* = 0.14	*p* = 0.007
III. Return to Supine						
Group 1 (*n* = 40)	110.2 [105.0; 117.3]	70.0 [67.1; 76.2]	64.7 [58.5; 70.0]	88.2 [79.8; 100.6]	6.3 [5.5; 7.2]	1085.1 [896.1; 1229.9]
Group 2 (*n* = 40)	117.2 [109.9; 122.9]	73.7 [69.1;79.1]	63.3 [59.9; 70.8]	84.9 [73.9; 93.3]	5.6 [4.8; 6.1]	1311.3 [1110.3; 1477.3]
Inter-Group Differences	*p* = 0.019	*p* = 0.1	*p* = 0.9	*p* = 0.23	*p* = 0.002	*p* < 0.001
Intra-Group Differences in Hemodynamic Parameters, According to Body Position						
Group 1	*pI-II* = 0.021 *pI-III* = 0.05 *pII-III* = 0.69	*pI-II* = 0.25 *pI-III* = 0.002 *pII-III* = 0.074	*pI-II* < 0.001 *pI-III* = 0.07 *pII-III* < 0.001	*pI-II* < 0.001 *pI-III* = 0.45 *pII-III* < 0.37	*pI-II* < 0.001 *pI-III* < 0.001 *pII-III* = 0.045	*pI-II* < 0.001 *pI-III* < 0.001 *pII-III* = 0.008
Group 2	*pI-II* < 0.001 *pI-III* = 0.006 *pII-III* = 0.0009	*pI-II* < 0.001 *pI-III* = 0.004 *pII-III* < 0.001	*pI-II* < 0.001 *pI-III* = 0.06 *pII-III* < 0.001	*pI-II* < 0.001 *pI-III* = 0.98 *pII-III* < 0.001	*pI-II* = 0.96 *pI-III* = 0.82 *pII-III* = 0.88	*pI-II* = 0.21 *pI-III* = 0.10 *pII-III* = 0.63

Note: data are presented as the median (Me) and interquartile range [25%; 75%] for parameters that were not normally distributed; Group 1, young European adults; Group 2, young African adults; *p*, inter-group differences; SBP, systolic blood pressure; DBP, diastolic blood pressure; HR, heart rate; SV, stroke volume; CO, cardiac output; TPR, total peripheral resistance.

The orthostatic hemodynamic changes were identical in both groups. During the HUTT, increases in SBP, DBP, HR, and TPR were observed, as well as minimal changes in CO. Different degrees of increase in the measured parameters were seen across the groups. In Group 2, there were significantly greater increases in SBP (*p* < 0.001), DBP (*p* = 0.014), and TPR (*p* = 0.007) during the HUTT; However, the inter-group differences in the other parameters were not significant.

The return to the horizontal position (return to supine) was accompanied by a recovery to the hemodynamic values of the initial baseline supine position, but there was no complete recovery for most parameters. In Group 2, hemodynamic recovery was faster after the passive orthostatic test with a standardized height for the hydrostatic column: of the six recorded hemodynamic parameters, four showed complete recovery (PI-III > 0.05) and SBP and DBP remained above their baseline values (PI-III = 0.006 and 0.004, respectively). In Group 1, of the six hemodynamic parameters, only HR and SV recovered to their baseline values (PI-III < 0.05).

The CAVI values for both racial groups were within the normal age range [25], but our comparative inter-group statistical analysis of the index revealed significant differences. The CAVI was significantly higher in the European group compared to the African group (5.38 [5.28;5.72] vs. 5.15 [4.66;5.44]; *p* = 0.03).

Our statistical analysis of the orthostatic changes in hemodynamic parameters only provided an average value and did not consider intra- and inter-individual differences. As such, the averaged results could infer incorrect information about the real orthostatic changes. The most significant indicator for a diagnosis of OH and OHT is ΔSBP. Averaging the indicators levels out individual differences because the ΔSBP can be either plus (increase) or minus (decrease). The personalized approach that was used in our work allowed us to divide each group into three categories based on changes in SBP, as classified in the work presented above [15]: “increase” for +5 < ΔSBP mmHg; “same” for ΔSBP ± 5 mmHg; “decrease” for −5 > ΔSBP mmHg.

### 3.2. Detection of Preclinical Orthostatic Abnormalities That Precede the Development of AH

Most of the participants were classified in the “same” group (28 (70%) in the European group and 26 (65%) in the African group), while there were 13 (32.5%) Africans and 8 (20%) Europeans in the “increase” group. The “decrease” group was the least numerous, with four (10%) European participants and one (2.5%) African participant [Figure 1]. Gender differences within the racial groups were not considered as the main aim of this work was to find out whether preclinical orthostatic disturbances could be detected in young healthy adults by means of the new HUTT protocol.

It was of great Interest to investigate the repeatability of the diagnosis of preclinical orthostatic disturbances in two separate studies. We conducted one study on 33 healthy subjects of different ages ranging from 20 to 66 years (M 29.5 [24;38]; women: 51.5%). A second study was conducted between 3 and 14 days after the first. A matching diagnosis was observed in 30 of the 33 subjects (90.9%). One subject had a change in diagnosis from orthostatic normotension (ONT) to OH and two subjects had a change in diagnosis from ONT to OHT during the repeat examination. A retrospective survey of these patients identified two possible reasons for the diagnosis changes: poor sleep on the eve of the study and flu.

## 4. Discussion

Before discussing the research findings, it is important to consider terminology issues that relate to orthostatic disturbances that precede AH. In the prototype study [15], the phenotypes of orthostatic disturbances in young healthy adults were classified into three groups: “drop”, a decrease in ΔSBP of more than −5 mmHg; “same”, ΔSBP changes between −5 and +5 mm Hg; “rise”, an increase in ΔSBP of more than + 5 mmHg. The total 8-year incidence of hypertension was 8.4% in the “decrease” group, 6.8% in the “same” group, and 12.4% in the “increase” group (*p* < 0.001).

Thus, the minimum risk of developing AH was found in the “same” group, while the maximum risk was almost twice as high in the “rise” group. When ΔSBP reached values of ±20 mmHg or more (OH and OHT), this became a CVD risk factor. These findings allowed OH and OHT that were associated with clinical disturbances and the development of CVDs to be defined as clinical OH or OHT. Asymptomatic orthostatic disturbances (“rise” and “drop”) can be defined as preclinical OHT and OH. Orthostatic changes with a ΔSBP of ±5 mmHg that were accompanied by a minimal risk of developing AH could be defined as orthostatic normotension (ONT) [22]. This terminology emphasizes the theoretical possibility that preclinical OH and OHT can potentially evolve into clinical disorders (OH and OHT).

This assumption has been indirectly confirmed by another large prospective study [16], in which some very important data were obtained: 1. in people over 45 years of age, preclinical and clinical OH are associated with incident hypertension; 2. it has been shown that changes in ΔSBP from ONT levels to the diagnostic boundaries of clinical OH are accompanied by a progressive increase in the risk of AH; 3. the progression of preclinical OH is accompanied not only by an increase in the risk of AH but also by an increase in intima-media thickness. These findings were consistent with those in the literature, which have shown that clinical OH is associated with increased vascular stiffness [26,27,28]. The exploration of the relationship between preclinical and clinical orthostatic abnormalities and the structural changes in vascular walls that result in increased vascular stiffness is very important as the latter is one of the greatest risk factors for CVDs [29,30,31,32]. This raises the question which is primary: preclinical orthostatic abnormalities or increased vascular stiffness (i.e. the egg and chicken problem). A partial answer to this question was obtained in our earlier study on young healthy Europeans under 27 years of age [21]. We showed that all subjects with and without orthostatic impairments had pulse wave velocities that were within the age norm. However, this study had the limitation of a small sample, so the conclusion was not statistically valid. A large sample study is planned to be conducted soon.

Another important point to be considered when analyzing preclinical orthostatic disturbances is the methodological aspects of detecting these alterations. In order to identify clinical orthostatic disturbances, an active standing protocol was applied [17,18]. Hemodynamic measurements were taken 2–3 min into the active standing test. It was shown that the blood pressure that was measured during the early stabilization phase of this test differed significantly from the value in the steady state phase (5–10 min). The active standing study was proposed to identify the personalized risk of orthostatic disorders (syncope) [13,33]. Determining a predisposition to syncope and identifying preclinical orthostatic disturbances are two different tasks. For the first task, a maximum gravity load that was proportional to the individual’s height was applied. The second task required the use of a standard gravity load to assess the specificity of orthostatic circulatory regulation, which determines the orthostatic stability of circulation. The state of the sympathetic baroreflex is of key importance [34,35,36,37] as when under orthostatic stress, it initiates the activation of the autonomic sympathetic system [38]. The state of the autonomic nervous system is important for the vascular system to adequately respond to the gravity challenge. Disturbances in this system inevitably lead to clinical orthostatic disturbances and adequate correction reduces the risk of recurrent syncope [39]. Adaptive orthostatic processes also include the activation of the renin–angiotensin–aldosterone system [40] and the release of catecholamines and vasopressin [41,42]. The degree of the activation of these pressor systems is directly proportional to the height of the hydrostatic column (pressure) [41]. The complex adaptive process with a significant neurohormonal shift during changes in body position allowed us to conclude that measurements should be taken during the steady state phase (5–10 min). Thus, orthostatic circulatory changes depended on orthostatic circulatory regulation and the subject’s height, thereby determining the value of the hydrostatic column. This value was variable and varied widely from 145 to 195 cm and affected orthostatic changes and the accuracy of the diagnosis of preclinical orthostatic disturbances. We eliminated the influence of this variable in the new HUTT protocol (Luanda) by setting an individual tilt angle to ensure a constant hydrostatic column height of 133 cm [21].

The aim of our study was to ascertain whether orthostatic abnormalities and their racial differences in young healthy African and European adults could be detected using the new HUTT protocol, which provides a standard gravity load regardless of individual differences in the subjects’ height. The hemodynamic parameters at rest in both groups were within the normal limits and did not differ between the African and European groups (Table 2), except for systolic blood pressure, cardiac output, and total vascular resistance, which were higher in the African group. In our study, these three parameters were characterized by an increase in SBP from the baseline value of 115.0 [111.5; 120.6] to 120.3 [112.4; 123.4] mmHg (*pI-II* < 0.001) in the African group. In the European group, the increase was less significant (from 109.2 [103.4;118.7] to 111.9 [108.2;119.6] mmHg; *p* = 0.21). The orthostatic increase in DBP was also significant in the African group (*pI-II* < 0.001), while it was slight in the European group (*pI-II* = 0.25). In both groups, there was a significant increase in HR and a decrease in SV (*pI-II* < 0.001). The baseline CO values were significantly higher (*pI-II* < 0.001) and the TPR was significantly lower in the European group (*pI-II* < 0.001) than in the African group. Orthostatic changes in these parameters (i.e., a decrease in CO and an increase in TPR) were only significant in the European group (*pI-II* < 0.001) and there were only slight changes in these parameters in the African group (*pI-II* = 0.96 and *pI-II* = 0.21, respectively). During the HUTT, the orthostatic changes in the hemodynamic parameters were identical to those presented in the literature [43,44,45]. A graded increase or decrease in tilt angle was accompanied by a proportional change in hemodynamic indices (except SBP) and the autonomic nervous system [46].

A group analysis of the orthostatic hemodynamic changes was not appropriate due to the multidirectional individual changes. One of the main diagnostic indicators of orthostatic disorders is SBP. When this indicator is averaged during a group analysis, its individual multidirectional values are averaged out, which makes it impossible to diagnose orthostatic abnormalities. In our study, the averaged ΔSBP was +5.3 [5.2;5.8] in the African group and ΔSBP = +2.7 [2.4;2.9] in the European group. Our personalized analysis of this indicator in the two racial groups allowed each group to be divided into three subgroups.

In our study, after adjusting for height differences using the individual tilt angles, the overall results for the two racial groups were 7.5% of the subjects had preclinical OH, 67% of the subjects had ONT, and 25.5% of the subjects had preclinical OHT. In the active standing test study presented above (12), preclinical OH was found in 26.6% of the African American and White American participants, ONT was found in 57.2% of the subjects, and preclinical OHT was found in 16.2% of the subjects. The difference in the results of these two studies could be due to the different populations of subjects: African Americans vs. Africans and White Americans vs. Europeans. Another reason could be the difference in study protocols: active standing test with significant variations in hydrostatic column height (height) and the HUTT with a standardized hydrostatic column height, providing the use of a standard gravity load. The difference in the results of the two studies could be due to the different populations of subjects: African Americans vs. Africans and White Americans vs. Europeans. Another reason could be the difference in study protocols: active standing test with significant variations in hydrostatic column height (height) and the HUTT with a standardized hydrostatic column height, providing the use of a standard gravity load. An important difference between the two protocols is that in our study, the parameters were recorded during the steady state period (5–10 min) and in the active standing protocol, they were measured during the early stabilization period (2–3 min).

The major novel finding of the present work was the identification of early preclinical orthostatic alterations that predicted AH using the HUTT protocol, which allowed for the study of orthostatic circulatory regulation (i.e., the main cause of AH) under standard gravity load conditions. Using this Luanda protocol, preclinical OHT was identified in 32.5% of the African participants and 20% of the European participants and preclinical OH was identified in 2.5% and 10% of the participants, respectively. The high prevalence of preclinical OHT in the African group (32.5%) correlated with the highest prevalence of AH in Africa [47], which indicated a possible pathogenetic link. These results were surprising because in the young African adults, vascular stiffness was significantly lower than that in the young European adults, according to CAVI (Table 1). With increasing age, the increase in vascular stiffness is more rapid in Africans than in Europeans [48,49]. This suggestion was supported by the results of the study presented above [16], which showed that preclinical AH is accompanied by an increased risk of AH and a proportional increase in vascular stiffness.

The critical importance of this problem necessitates further research on early AH predictors to form the basis of new effective primary prevention strategies.

Yearly screening to identify preclinical orthostatic disturbances and vascular stiffness could be of great importance. The good repeatability of the diagnosis of these abnormalities could allow for the new HUTT protocol to be used in prospective cohort studies. The critical issues of the AH pandemic requires further research to form effective preventive measures and to accurately predict AH.

## 5. Conclusions

The results of this study showed the feasibility of the detection of preclinical orthostatic disturbances in young adults and the detection of their racial differences using the HUTT protocol, providing the use of a standard gravity load. Further study on the evolution of preclinical orthostatic disturbances and their relation to increasing vascular stiffness is necessary among large samples.

## 6. Perspectives

The results of this cross-sectional cohort study could provide a foundation for further prospective studies to elucidate the evolution of preclinical orthostatic disturbances and arterial stiffness, their transformation into risk factors (clinical OH and OHT), and vascular wall remodeling that leads to increased stiffness. The new HUTT protocol could be used in large scale multicenter studies to assess the orthostatic regulation of circulation, evaluate the individual rates of vascular ageing, and detect early AH predictors in populations with different ages, races, and geographic locations.

## Figures and Tables

**Figure 1 biomedicines-10-02156-f001:**
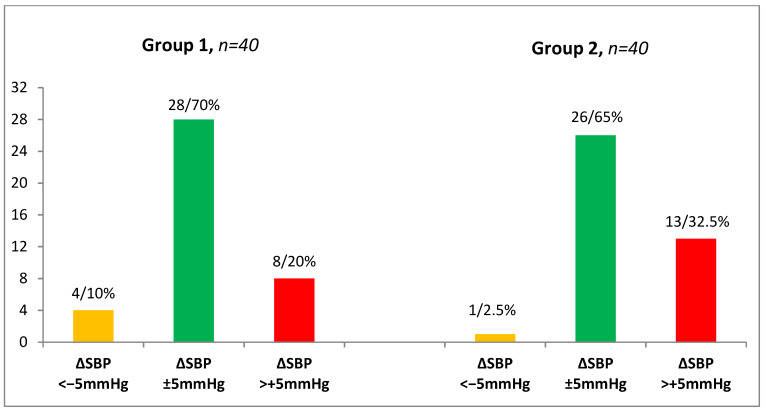
The personalized analysis of ΔSBP in different racial groups of young healthy adults (European participants, Group 1; African participants, Group 2) using a passive orthostatic test with a standardized hydrostatic column height. Note: orange column, “decrease” category (ΔSBP < −5 mmHg); green column, “same” category (ΔSBP ± 5 mmHg); red column, “increase” category (ΔSBP > +5 mmHg); *Y*-axis, the number of subjects.

**Table 1 biomedicines-10-02156-t001:** The characteristics of the European and African young adults.

	Group 1 *n =* 40	Group 2 *n =* 40	*p*
Age (Years)	22.3 ± 1.24	21 ± 1.3	0.54
Sex (Male/Female)	22/18	24/16	
Height (cm)	172.3 ± 9.3	167.5 ± 7.5	0.06
Weight (kg)	69.1 ± 8.3	65.2 ± 12.2	0.72
Body Mass Index (kg/m^2^)	23.3 ± 3.4	22.3 ± 4.9	0.6
CAVI	5.38 [5.28; 5.72]	5.15 [4.66; 5.44]	0.03

Note: indicators are presented as the arithmetic mean (M) ± standard deviation (SD); cardio-ankle vascular index (CAVI) is presented as the median (Me) and interquartile range [25%; 75%]; Group 1, young European adults; Group 2, young African adults; *p*, inter-group differences.

## Data Availability

The data that support the findings of this study are available from the corresponding author, Victor N. Dorogovtsev, upon reasonable request.

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
