# Peer review of "Detection of Preclinical Orthostatic Disorders in Young African and European Adults Using the Head-Up Tilt Test with a Standardized Hydrostatic Column Height: A Pilot Study"

_biomedicines, 2022, doi:10.3390/biomedicines10092156_

Round 1

Reviewer 1 Report

Performing this research is of great importance and of interest to the scientific and medical community. The methods are of practical value as these are non-invasive and can be done at relative low expense.

As this is a methodological paper, the paper would be of significantly greater value of repeated measurements/repeatability indicators (repeated observations done at two separate occassions) in subjects could be presented. 

It is also suggested the authors contact a native English speaker to shorten, smoothen and simplify the text.  Much of the text content can also be presented in the Tables. 

The results describing the differences between groups are of interest but must be supported by repeatability data; the graphs look fine.

Author Response

Dear reviewer,
Thank you for reviewing our article, we have tried to clear your doubts and queries. We have also made changes as per your comments on the paper. 

Performing this research is of great importance and of interest to the scientific and medical community. The methods are of practical value as these are non-invasive and can be done at relative low expense.

  1. As this is a methodological paper, the paper would be of significantly greater value of repeated measurements/repeatability indicators (repeated observations done at two separate occassions) in subjects could be presented. 

This question came up during the development of the new HUTT protocol. We conducted such a study on 33 healthy subjects of different age groups. The repeatability of the diagnosis of preclinical orthostatic abnormalities at two separate occasions was above 90%. Repeatability information has been added to the results section.

  1. It is also suggested the authors contact a native English speaker to shorten, smoothen and simplify the text.  Much of the text content can also be presented in the Tables. 

We have tried to take your comments about the English language into account. Significant changes have been made to the text. The data on vascular stiffness presented in the discussion have been added to Table 1.

The results describing the differences between groups are of interest but must be supported by repeatability data; the graphs look fine.

Reviewer 2 Report

The manuscript entitled "Detection of preclinical orthostatic disorders with the head-up tilt test standardized on hydrostatic column height in African and European young adults: a Pilot study" seems interesting, properly designed regarding experimental analysis, well-written and the results adequately presented. In general, some minor questions and suggestions are made below and should be addressed before considering its publication in Biomedicines journal: 

1. In introduction, please consider discussing in more detail the possible relationship between AH and different races as this seems a possible influencing factor. 

2. Was there a specific reason for the study have been performed in African and European adults? Was it to study the race as a possible influencing factor in both populations? And why in young people? Please clarifty this in the manuscript.

 3. Due to the high number of abbreviations, it would be a good option to do a brief list of the most common abbreviations used throughout the manuscript.

4. The relevance of the study on the health research field should be highlighted in the manuscript as well as its possible benefits for practical applications.

5. Format "p" regarding probability or p values in italic letter. Please revise it along the manuscript, including subtitles, tables and figures.

6. The statistical analysis is missing in Figure 1.

7. The main advantages and disadvantages of HUTT protocol should be discussed in more detail in the manuscript.

Minor comments:

- Abstract: Please consult the journal guidelines for the abstract preparation.

- Line 16: Place "HUTT" within parentheses after the full name as done above for AH (line 13).

- Line 38: "It" is repeated a lot in the previous sentences. Please replace it here in the beginning of the sentence.

- Lines 44-45: Please consider revising the verbs by replacing by -ing form according to sentence, namely "engaging", "maintaining", "limiting", "reducing" and "consuming". 

- Line 45: Add "and" before "smoking".

- Lines 52-53: Use just "CVD" here since this abbreviation is already defined above.

- Line 61: Delete the extra space before the beginning of the sentence.

- Lines 74-76: The information in this paragraph seems repeated with the one presented in the previous paragraph. Please revise it.

- Line 79: In this sentence, the age of the subjects under study is cited as "20-25 years". However, in the abstract, it is stated "20-23 years". Please revise it.

- Line 86: Add "and" to replace the comma ("primary and secondary").

- Line 87: Use the plural "dysfunctions".

- Lines 96-97: Add a space before line 97 and the footnotes of the Table 1.

- Line 100: Consider using the full name of "HUTT" in the subtitle 2.2.

- Line 115: Please indicate the full name of abbreviation "BP".

- Line 116: Consider adding "while" before "HR".

- Lines 118-120: Please revise carefully the subtitle 2.4.

- Line 122: Format "®" above the line "(Stat Soft®)". 

- Lines 134-135: Please revise the subtitle 3.1. and use this sentence as the first sentence of this subsection to indicate the content of Table 2.

- Consult the journal guidelines for the presentation of subtitles, tables and figures including the ones regarding the proper format.

- Table 2: Please correct the content "pI-III=0.h004". 

- Lines 141-142: A space is missing between these two lines.

- Line 152: Delete the point before "(Table 2)". Also, add "while the" before "other".

- Figure 1: Please identify the axis y (for example, as "Number of subjects"). Also, consider placing the percentages above each column within parentheses. Replace the comma in the number by a point ("2.5%").

- Line 187: Add "were" before "28". 

- Line 188: Add "while" before "there".

- Line 190: Replace "adult" by the plural "adults".

- Line 195: Format properly the reference [15].

- Line 212: Delete the extra space between "2." and "It...".

- Line 225: Please consider deleting the abbreviation "PWV" since it seems that is not used in any other sentence throughout the manuscript. In this sentence, the authors stated that the limitation of the study cited is "a small sample". How small is the sample? Please include this information in this sentence.

- Line 251: Delete the point after "cm" and before the reference [19].

- Lines 317-326: Please consider formatting the information of this subsection numbered "6. Perspectives" in just one paragraph.

- Lines 328-335: Consult the journal guidelines and use just the abbreviations of authors' names (defined above in page 1) in this subsection regarding "Author contributions".

- Lines 337-339: Please indicate the country and city where the clinical procedure was performed.

- References: Please format all the references following the journal guidelines.

Author Response

Dear reviewer,
Thank you for reviewing our article, we have tried to clear your doubts and queries. We have also made changes as per your comments on the paper. Please see the attachment below for our responses. 

Reviewer 3 Report

The work submitted for publication in BIOMEDICINES entitled: “ Detection of preclinical orthostatic disorders with the head-up tilt test standardized on hydrostatic column height in African and European young adults: a Pilot study ” by Victor N. Dorogovtsev, Dmitry Yankevich, Marina V. Petrova, Vladimir Troshin, Aleksander E. Severin, Ilya V. Borisov, Julia A. Podolskaya, Andrey V. Grechko is a case analysis of the common arterial hypertension in the population, trying to detect the factors that generate it (if possible). Actually, the approach of dealing with young people is appealing, but maybe too young still. And the solutions based on preclinical orthostatic disturbances could also be controversial.

Some concerns:

-Which was the reason of the choice of the people tested? And the age? Probably the dimension is due to the economical resources but this should be clarified, because 40 people is not a high number.

-How the diet in both regions affect?

-Why the list the references is rather short?

-Then, one of the conclusions states that it is possible to see the disturbances in young people, but how we can be sure that those then generate AH?

If the above concerns are properly addressed in the text, with some discussion, the paper could be considered for publication.

Author Response

Dear reviewer, 
Thank you for reviewing our paper. Our responses to your very useful comments are written below. 

  1. Which was the reason of the choice of the people tested? And the age? Probably the dimension is due to the economical resources but this should be clarified, because 40 people is not a high number.

The main reason for the choice of young subjects is the detection of preclinical orthostatic abnormalities, which increase the risk of arterial hypertension. Such abnormalities are associated with an accelerated increase in vascular stiffness, suggesting a possible pathogenetic link between orthostatic hemodynamic abnormalities and cardiovascular disease. Our pilot study was carried out on a small sample, but even with these limitations, the new protocol identified an increased risk of arterial hypertension in a proportion of young, healthy adults. The results have led to the preparation of a prospective cohort study in a large sample of young healthy subjects.

  1. How the diet in both regions affect?

Studying the effect of European and African dietary patterns on the development of hypertension is important, but this was not the aim of this study. Such an analysis would have been difficult due to the 2-year stay far from their country of origin of African students from different African countries.

  1. Why the list the references is rather short?

The list of references is indeed small because our study analyses the possibility of diagnosing early preclinical abnormalities (predictors) that have not yet emerged from the experimental clinical category.

With the application of evidence-based medicine it is certain that there will be many more publications on this topic. This is only the beginning!

  1. Then, one of the conclusions states that it is possible to see the disturbances in young people, but how we can be sure that those then generate AH?

This was shown in the representative prospective 8-year CARDIA Study. The active standing test was used in this study. The disadvantages of this test are discussed in our paper. We have proposed a specific test to detect preclinical asymptomatic orthostatic disturbances. Such a test carries no risk of syncope and provides a standard gravity load regardless of subjects' height.

Round 2

Reviewer 1 Report

A well revised version. As is I have no further comments.

Author Response

Dear reviewer,
Thank you for the revision and acceptance of our revised version.